# Analyzing real-world adverse events of spironolactone with the FAERS database

Guowei Lin[1◉], Ru Chen[2◉], Chaoning Wen[1], Zhuopin Li[1], Xiangshan Yan[1], Lixian Li[1]*

**1** Department of Urology, Putian University Affiliated Hospital, Putian, Fujian, China, **2** Department of Urology, Fujian Medical University Union Hospital, Fuzhou, Fujian, China

◉ These authors contributed equally to this work.
* lilixian007@sina.com

## Abstract

### Background

Spironolactone, a potassium-sparing diuretic, is commonly prescribed for conditions such as heart failure, hypertension, and hyperaldosteronism. This study aims to explore and analyze the safety profile of spironolactone by examining adverse event reports from the FDA Adverse Event Reporting System (FAERS) database.

### Methods

This study conducted a retrospective pharmacovigilance analysis using FAERS data (2004 Q1–2024 Q3). Adverse drug events (ADEs) related to spironolactone were identified and categorized by system organ class and specific adverse events. Statistical methods such as Proportional Reporting Ratio (PRR), Reporting Odds Ratio (ROR), Bayesian Confidence Propagation Neural Network (BCPNN), and Empirical Bayesian Geometric Mean (EBGM) were employed to detect potential safety signals.

### Results

A total of 8,566 ADE reports were associated with spironolactone, with 2409 preferred terms and 25 system organ classes showing significant disproportionality. Notable rare ADEs identified included endometriosis male (n = 7; ROR 13615.84), 5-alpha-reductase deficiency (n = 5; ROR 1620.81), bulbospinal muscular atrophy congenital (n = 6; ROR 402.42) and double-hit lymphoma (n = 5; ROR 243.12).

### Conclusion

While most findings were consistent with spironolactone's known effects, new signals, including a potential link to male endometriosis in high-risk groups, were identified. Further research is needed to confirm these findings and improve long-term safety assessment and clinical management.

**Data availability statement:** The raw FAERS data used in this study are publicly available from the U.S. FDA (https://fis.fda.gov/extensions/FPD-QDE-FAERS/FPD-QDE-FAERS.html). The processed dataset and analysis scripts generated during the current study are available from the original figures and tables, as well as the Mendeley Data repository (https://doi.org/10.17632/fpfknkmz5h.1).

**Funding:** The author(s) received no specific funding for this work.

**Competing interests:** The authors have no relevant affiliations or financial involvement with any organization or entity with a financial interest in or financial conflict with the subject matter or materials discussed in the manuscript. This includes employment, consultancies, honoraria, stock ownership or options, expert testimony, grants or patents received or pending, or royalties.

## 1. Introduction

Spironolactone is the first mineralocorticoid receptor antagonist developed and is widely used in the treatment of hypertension, primary hyperaldosteronism, and peripheral edema associated with heart failure, as well as other conditions related to aldosteronism [1]. As a synthetic 17-lactone, spironolactone is a non-selective mineralocorticoid receptor antagonist that has been in clinical use since its introduction in 1960, maintaining its relevance for over six decades [2,3]. In addition to its use in managing hypertension, spironolactone is also employed in the treatment of acne in women due to its antagonistic effects on both progesterone and androgen receptors [4,5]. Recently, spironolactone has gained attention for the treatment of resistant hypertension [6]. A study demonstrated that when added to regimens including diuretics or angiotensin-converting enzyme inhibitors, spironolactone significantly improved blood pressure control after 6 weeks and 6 months of treatment [7]. Due to its multifaceted therapeutic effects and favorable tolerability, spironolactone remains a commonly used drug in clinical practice, particularly in the management of hypertension, heart failure, and endocrine-related disorders.

Despite the therapeutic benefits of spironolactone, it is associated with a range of adverse drug events (ADEs). The common ADE is menstrual irregularities (15%−30%), which are dose-dependent and can be alleviated with oral contraceptives or intrauterine devices [8,9]. Other infrequent ADEs include increased urination, lightheadedness, headaches, nausea, vomiting, breast discomfort, and breast enlargement [10]. As a potassium-retaining diuretic, elevated potassium levels (hyperkalemia) represent another possible side effect, especially in individuals with renal dysfunction or heart failure, and its occurrence is more common at higher dosages [11]. According to the prescribing information, routine monitoring of potassium levels is recommended for patients receiving spironolactone for Food and Drug Administration (FDA)-approved indications, including hypertension and heart failure. Additionally, there are reports of rare but severe adverse reactions. One study reported a case of granulocytopenia in a patient with cirrhotic ascites who was treated with spironolactone [12]. A case report described a male patient who developed a lupus-like skin reaction following spironolactone use [13]. Recognizing and delineating these ADEs is essential for enhancing patient safety and facilitating well-informed clinical choices.

In contrast to laboratory studies and clinical trials, pharmacovigilance information offers a more accurate representation of real-world drug usage and is essential for post-market surveillance [14]. The FDA Adverse Event Reporting System (FAERS) is the publicly accessible repository for voluntarily reported ADEs, gathering information from healthcare providers, consumers, manufacturers, and other stakeholders. It plays a crucial role in notifying healthcare providers and the general public about possible drug-related hazards [15,16]. Data mining techniques, including the Reporting Odds Ratio (ROR), Proportional Reporting Ratio (PRR), Bayesian Confidence Propagation Neural Network (BCPNN) and the empirical Bayesian geometric mean (EBGM) are widely used for signal detection in pharmacovigilance databases [17,18]. These approaches apply statistical methods to uncover associations or unanticipated

events within extensive datasets. Many studies have utilized the FAERS database to investigate ADEs of various medications [19–24]. However, to date, no research has utilized the FAERS database to analyze the adverse reactions associated with spironolactone. The objective of this study is to thoroughly examine the risk of adverse reactions associated with spironolactone use through FAERS and to identify potential adverse reactions not specified in the drug labeling, providing valuable insights for clinical practice.

## 2. Methods

### 2.1 Data collection

The FAERS database is a spontaneous reporting system that gathers ADE reports from a wide range of sources, including patients, healthcare providers, and pharmaceutical companies. It supports post-market surveillance and aims to enhance public health by aggregating and analyzing safety data, which is particularly beneficial for managing long-term conditions [25]. In this study, we conducted a retrospective pharmacovigilance analysis using FAERS data from the first quarter (Q1) of 2004 to the third quarter (Q3) of 2024. We also extracted patient demographic and clinical details, such as sex, age, reporting region, type of reporter, date of report, and outcomes associated with spironolactone-related ADEs.

### 2.2 Extracting and analyzing data

In accordance with the Medical Dictionary for Regulatory Activities (MedDRA) (Version 26.1), ADEs were classified using Preferred Terms (PTs) from the FAERS database. Each PT can be linked to multiple High-Level Terms, High-Level Group Terms, and System Organ Classes (SOCs). Within the FAERS database, a single ADE case may be reported multiple times, resulting in several entries for the same patient. Therefore, data cleaning procedures were implemented prior to analysis. Specifically, duplicate entries were identified and removed using unique identifiers in FAERS, such as the CASEID (which uniquely identifies each case) and ISR (Individual Safety Report) number (which tracks updates to the same case). For example, if multiple reports shared the same CASEID but different ISR numbers (indicating updates or revisions), only the report with the latest submission date (i.e., the most recent ISR version) was retained. This process ensured that each case was represented by its most updated clinical information, eliminating redundant data from initial reports and subsequent amendments. It is important to note that drug names in FAERS are typically reported in free-text format, which may include both generic and brand names, as well as research codes, with potential for spelling inconsistencies. To address this, a comprehensive drug name reference was applied, encompassing all known generic names, brand names, and study codes for FDA-approved spironolactone.

PRR, ROR, BCPNN, and EBGM [18,26–28] are commonly employed techniques in pharmacovigilance for detecting ADE signals. While PRR measures relative risk, it tends to be highly sensitive and vulnerable to false positives, particularly when the number of reported incidents is small. In contrast, ROR offers more reliable estimates of risk or hazard ratios, with smaller deviations compared to other methods, and we applied a threshold of ROR ≥ 3 and 95% CI lower limit > 1 as recommended by Rothman et al [18]. BCPNN is recognized for its robustness, even with a limited number of reports, using an information component lower limit (IC025) > 0 as validated by Bate et al [27], while EBGM is particularly effective at identifying signals for rare events with a threshold of EBGM05 > 2 following Dumouchel [28]. For PRR, we adopted a criterion of PRR ≥ 2 and 95% CI lower limit > 1 as described by Evans et al [26]. This study integrated these methods to broaden detection scope and validate findings across statistical frameworks. A signal was deemed significant only when all four algorithms met their thresholds (e.g., ROR ≥ 3, PRR ≥ 2, IC025 > 0, EBGM05 > 2), a strategy designed to minimize false positives through cross-validation. Discordant results—where one method met criteria but others did not—were not prioritized to ensure consistency in identifying genuine associations. Detailed workflows for signal detection, including 2 × 2 contingency table construction and threshold evaluations, are provided in S1 Table, enhancing methodological transparency. The combined approach ensures reliable identification of potential rare ADEs

by balancing sensitivity and specificity. The combined use of these algorithms ensures cross-validation, minimizes false positive rates, and enhances the identification of potential rare ADEs by adjusting thresholds and variances. Each method utilizes a 2x2 contingency table (Table 1), with detailed formulas and cutoff values presented in Table 2. A higher value indicates a stronger signal, suggesting a more significant association between the drug of interest and the ADEs. Ultimately, effective ADEs detection requires the fulfillment of positive signal criteria set by the four algorithms mentioned. Data related to spironolactone were meticulously processed and statistically analyzed using tools such as Excel and R Studio (Version 4.3.1, https://www.r-project.org/). The overall research approach is depicted in the flow-chart (**Fig 1**).

**Table 1. Four grid table.**

|  | Spironolactone-related ADEs | Non-Spironolactone-related ADEs | Total |
|---|---|---|---|
| Spironolactone | a | b | a+b |
| Non-Spironolactone | c | d | c+d |
| Total | a+c | b+d | N=a+b+c+d |

Abbreviations: ADEs, adverse drug events.

**Table 2. ROR, PRR, BCPNN, and EBGM methods, formulas, and thresholds.**

| Method | Formula | Threshold |
|---|---|---|
| ROR | $ROR = \frac{a \ / \ c}{b \ / \ d}$ <br><br> $SE(\ln ROR) = \sqrt{\frac{1}{a} + \frac{1}{b} + \frac{1}{c} + \frac{1}{d}}$ <br><br> $95\%CI = e^{\ln(ROR) \pm 1.96se}$ | $a \geq 3$ <br> $ROR \geq 3$ <br> 95%CI (lower limit) > 1 |
| PRR | $PRR = \frac{a \ / \ (a+b)}{c \ / \ (c+d)}$ <br><br> $SE(\ln PRR) = \sqrt{\frac{1}{a} - \frac{1}{a+b} + \frac{1}{c} - \frac{1}{c+d}}$ <br><br> $95\%CI = e^{\ln(PRR) \pm 1.96se}$ | $a \geq 3$ <br> $PRR \geq 2$ <br> 95%CI (lower limit) > 1 |
| BCPNN | $IC = \log_2 \frac{p(x, y)}{p(x)p(y)} = \log_2 \frac{a(a+b+c+d)}{(a+b)(a+c)}$ <br><br> $E(IC) = \log_2 \frac{(a+\gamma 11)(a+b+c+d+\alpha)(a+b+c+d+\beta)}{(a+b+c+d+\gamma)(a+b+\alpha 1)(a+c+\beta 1)}$ <br><br> $V(IC) = \frac{1}{(\ln 2)^2} \left[ \frac{(a+b+c+d)-a+\gamma-\gamma 11}{(a+\gamma 11)(1+a+b+c+d+\gamma)} + \frac{(a+b+c+d)-(a+b)+a-\alpha 1}{(a+b+\alpha 1)(1+a+b+c+d+\alpha)} + \frac{(a+b+c+d+\alpha)-(a+c)+\beta-\beta 1}{(a+b+\beta 1)(1+a+b+c+d+\beta)} \right]$ <br><br> $\gamma = \gamma 11 \frac{(a+b+c+d+\alpha)(a+b+c+d+\beta)}{(a+b+\alpha 1)(a+c+\beta 1)}$ <br><br> $IC - 2SD = E(IC) - 2\sqrt{V(IC)}$ | IC025 > 0 |
| EBGM | $EBGM = \frac{a(a+b+c+d)}{(a+c)(a+b)}$ <br><br> $SE(\ln EBGM) = \sqrt{\frac{1}{a} + \frac{1}{b} + \frac{1}{c} + \frac{1}{d}}$ <br><br> $95\%CI = e^{\ln(EBGM) \pm 1.96se}$ | EBGM05 > 2 |

Abbreviations: ROR, Reporting Odds Ratio; PRR, Proportional Reporting Ratio; BCPNN, Bayesian Confidence Propagation Neural Network; EBGM, Empirical Bayesian Geometric Mean; SE, Standard Error; CI, Confidence Interval; IC, Information Component.

 

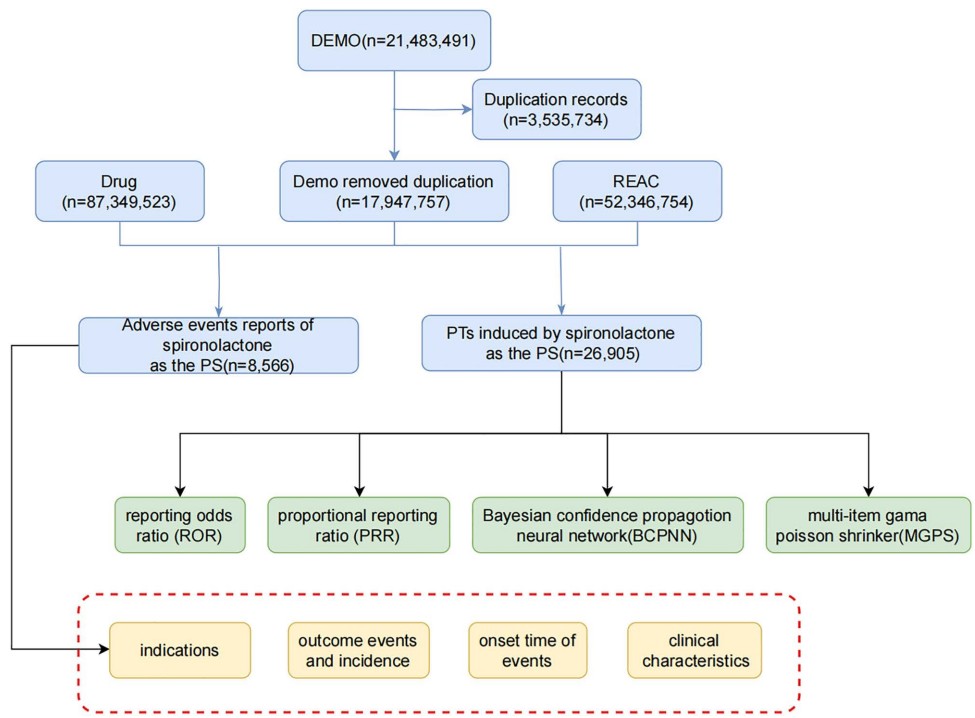

**Fig 1. Flow chart.** The flow diagram of selecting spironolactone-related ADEs from FAERS database. Abbreviations: DEMO, demographic; REAC, Reporting Event Adverse Condition; PTs, Preferred Terms; PS, Primary Suspect; ROR, Reporting Odds Ratio; PRR, Proportional Reporting Ratio; BCPNN, Bayesian Confidence Propagation Neural Network; MGPS, multi-item gamma poisson shrinker.

## 3. Results

### 3.1 ADE reports and clinical data

A total of 17,947,757 ADE reports were recorded in the FAERS database from Q1 2004 to Q3 2024 (**Fig 2**). Of these, 8,566 reports listed spironolactone as the main suspect drug, covering 2,409 PTs. A greater number of reports were submitted by females (4,292 reports, 50.11%) compared to males (3,542 reports, 41.35%). The age group most frequently affected was individuals aged 75 years and older (2,653 reports, 30.97%). Physicians contributed the largest proportion of reports (2,580 reports, 30.12%), with the United States being the leading country for case submissions (2,441 reports, 28.50%). Regarding clinical outcomes, the most frequent result was hospitalization (3,954 reports, 44.39%), followed by other serious outcomes (3,421 reports, 38.41%). Further information is available in **Table 3**.

### 3.2 Signals detection associated with spironolactone

**3.2.1. Signal detection at the level of SOCs.** Signal detection was performed at the SOC level. Based on statistical analysis, spironolactone-related ADEs primarily impacted 25 distinct SOCs. Among these, the SOC with the highest frequency of ADEs was "general disorders and administration site conditions" (n = 3,555, ROR 0.7, PRR 0.74, IC −0.44, EBGM 0.74). However, the "metabolism and nutrition disorders" (n = 3,234, ROR 6.06, PRR 5.45, IC 2.44, EBGM 5.44) and "reproductive system and breast disorders" (n = 913, ROR 4.11, PRR 4.01, IC 2, EBGM 4) showed the highest ROR values, suggesting robust correlations across all four statistical methods. Several of these findings were consistent with well-known ADEs mentioned in the drug labeling, further reinforcing the credibility of the data. Notably, certain SOCs were linked to significant ADEs not previously documented in the drug inserts, including "congenital, familial and genetic

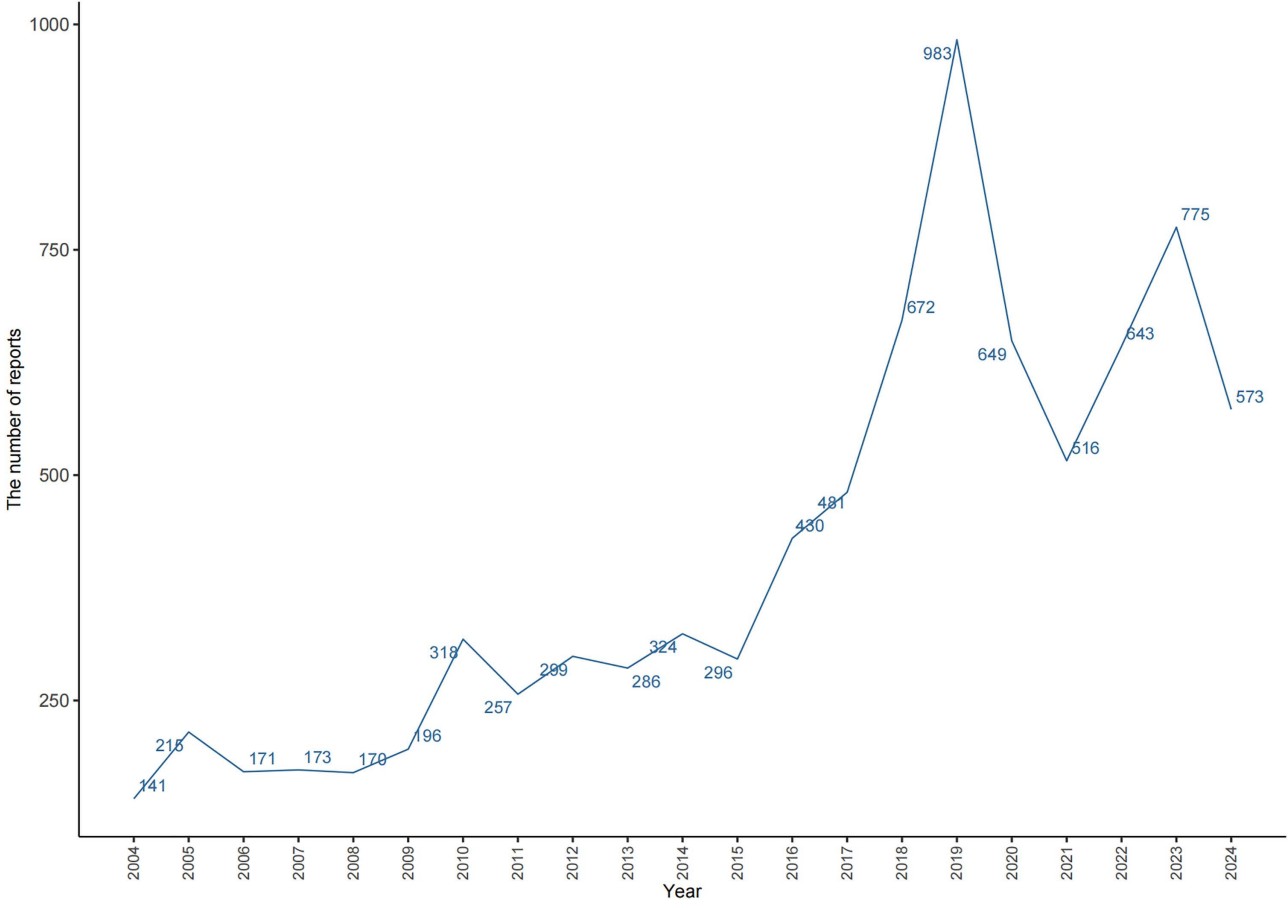

**Fig 2. Year curve chart.** Curve plot of the adverse reaction reports for spironolactone over the years.

disorders" (n = 160, ROR 1.88, PRR 1.87, IC 0.9, EBGM 1.87), "ear and labyrinth disorders" (n = 141, ROR 1.18, PRR 1.18, IC 0.24, EBGM 1.18), and "pregnancy, puerperium, and perinatal conditions" (n = 86, ROR 0.72, PRR 0.72, IC −0.47, EBGM 0.72) (Table 4).

 **3.2.2 ADEs frequency analysis.** Table 5 presents the top 30 ADEs associated with spironolactone, ranked according to their signal strength. Notably, hyperkalemia (n = 1,308; ROR 90.35, PRR 86.01, IC 6.36, EBGM 82.41) and nipple pain (n = 42; ROR 46.19, PRR 46.12, IC 5.49, EBGM 45.07) exhibited significantly higher frequency rates and signal magnitudes. Additionally, ADEs such as male endometriosis (n = 7; ROR 13,615.84, PRR 13,612.3, IC 10.73, EBGM 1,702.41), 5-alpha-reductase deficiency (n = 5; ROR 1,620.81, PRR 1,620.51, IC 9.79, EBGM 884.37), congenital bulbospinal muscular atrophy (n = 6; ROR 402.42, PRR 402.33, IC 8.38, EBGM 333.53), and double-hit lymphoma (n = 5; ROR 243.12, PRR 243.08, IC 7.76, EBGM 216.18) were recognized as potential new ADE signals that were not included in the drug's labeling.

## 4. Discussion

Spironolactone, a potassium-sparing diuretic and mineralocorticoid receptor antagonist, is commonly used to manage a range of conditions, including cirrhosis [29], non-alcoholic fatty liver disease [30], polycystic ovary syndrome [31], diabetic nephropathy [32], and hypertension [33]. According to the most recent prescribing information from 2018, potassium levels should be monitored within one week of initiating or titrating spironolactone, then monthly for the first three months, quarterly

**Table 3.  Basic information on ADEs related to spironolactone from the FAERS database.**

| Variable | Total |
|---|---|
| Number of events | 8566 |
| Sex | |
| Female | 4292(50.11) |
| Male | 3542(41.35) |
| Unknown | 732(8.55) |
| Age | |
| <65 | 2472(28.86) |
| 65~75 | 1666(19.45) |
| >=75 | 2653(30.97) |
| Unknow | 1775(20.72) |
| Reporter | |
| Physician | 2580(30.12) |
| Pharmacist | 2247(26.23) |
| Consumer | 2129(24.85) |
| Other health-professional | 1341(15.65) |
| Unknown | 269(3.15) |
| Reported countries | |
| Other | 3678(42.93) |
| United States | 2441(28.50) |
| France | 1379(16.10) |
| United Kingdom | 716(8.36) |
| Germany | 352(4.11) |
| Outcomes | |
| Hospitalization | 3954(44.39) |
| Other serious | 3421(38.41) |
| Life threatening | 660(7.41) |
| Death | 498(5.59) |
| Disability | 202(2.27) |
| Required intervention to Prevent Permanent Impairment/Damage | 128(1.44) |
| Congenital anomaly | 44(0.49) |

Abbreviations: ADEs, adverse drug events; FAERS, FDA Adverse Event Reporting System.

for the following year, and subsequently every six months [34]. In addition to its approved indications, spironolactone is widely prescribed off-label for acne treatment due to its effectiveness, affordability, and favorable side effect profile [35,36]. However, recommendations for potassium monitoring are not provided for this off-label use. Spironolactone is commonly regarded as a safe and well-tolerated medication, but an observational study involving 788 hospitalized patients reported several adverse reactions. The most frequent ADE was hyperkalemia, noted in 68 out of 788 patients (8.6%), followed by dehydration (3.4%), hyponatremia (2.4%), gastrointestinal symptoms (2.3%), including appetite loss, nausea, vomiting, and diarrhea, and neurological disorders (2.0%), such as headache, dizziness, tremors, and confusion. Skin rashes and male breast enlargement were observed in 10 patients (1.3%), while other ADEs were less commonly reported [37]. Given its more than 60 years of market use, continued monitoring of spironolactone's clinical application and its associated ADEs is essential for ensuring both safety and efficacy. This study aims to comprehensively assess spironolactone-related adverse reactions by analyzing data from the FAERS database, spanning from the first quarter of 2004 to the third quarter of 2024.

Table 4. The signal strength of ADEs of spironolactone at the SOC level in FAERS database.

| System organ class | Case Reports | ROR(95% CI) | PRR(95% CI) | χ2 | IC(IC025) | EBGM(EBGM05) |
|---|---|---|---|---|---|---|
| metabolism and nutrition disorders | 3234 | 6.06(5.84, 6.28) * | 5.45(5.24, 5.67) * | 11982.07 | 2.44(2.39) * | 5.44(5.27) * |
| reproductive system and breast disorders | 913 | 4.11(3.85, 4.39) * | 4.01(3.78, 4.25) * | 2073.34 | 2(1.9) * | 4(3.79) * |
| renal and urinary disorders | 1701 | 3.5(3.33, 3.68) * | 3.34(3.21, 3.47) * | 2839.46 | 1.74(1.67) * | 3.34(3.2) * |
| cardiac disorders | 1551 | 2.17(2.06, 2.28) | 2.1(2.02, 2.18) * | 919.63 | 1.07(1) * | 2.1(2.01) * |
| endocrine disorders | 143 | 2.04(1.73, 2.41) | 2.04(1.74, 2.39) * | 75.52 | 1.03(0.79) * | 2.04(1.77) |
| congenital, familial and genetic disorders | 160 | 1.88(1.61, 2.19) | 1.87(1.6, 2.19) | 64.96 | 0.9(0.68) * | 1.87(1.64) |
| vascular disorders | 942 | 1.59(1.49, 1.7) | 1.57(1.48, 1.67) | 201.57 | 0.65(0.56) * | 1.57(1.49) |
| hepatobiliary disorders | 344 | 1.36(1.22, 1.52) | 1.36(1.23, 1.5) | 32.68 | 0.44(0.29) * | 1.36(1.24) |
| immune system disorders | 415 | 1.36(1.23, 1.49) | 1.35(1.22, 1.49) | 38.26 | 0.43(0.29) * | 1.35(1.25) |
| investigations | 2188 | 1.3(1.24, 1.36) | 1.27(1.22, 1.32) | 137.95 | 0.35(0.29) * | 1.27(1.23) |
| ear and labyrinth disorders | 141 | 1.18(1, 1.39) | 1.18(1.01, 1.38) | 3.92 | 0.24(0) * | 1.18(1.03) |
| skin and subcutaneous tissue disorders | 1543 | 1.04(0.99, 1.1) | 1.04(1, 1.08) | 2.27 | 0.05(−0.02) | 1.04(0.99) |
| blood and lymphatic system disorders | 436 | 0.92(0.84, 1.01) | 0.92(0.83, 1.01) | 2.79 | −0.11(−0.25) | 0.92(0.85) |
| nervous system disorders | 2021 | 0.85(0.81, 0.89) | 0.86(0.83, 0.89) | 50.52 | −0.22(−0.28) | 0.86(0.83) |
| gastrointestinal disorders | 1991 | 0.83(0.79, 0.87) | 0.84(0.81, 0.87) | 65.88 | −0.25(−0.32) | 0.84(0.81) |
| respiratory, thoracic and mediastinal disorders | 1022 | 0.76(0.72, 0.81) | 0.77(0.73, 0.82) | 71.28 | −0.37(−0.46) | 0.77(0.73) |
| pregnancy, puerperium and perinatal conditions | 86 | 0.72(0.58, 0.89) | 0.72(0.58, 0.89) | 9.35 | −0.47(−0.78) | 0.72(0.6) |
| general disorders and administration site conditions | 3555 | 0.7(0.67, 0.72) | 0.74(0.71, 0.77) | 403.36 | −0.44(−0.49) | 0.74(0.72) |
| injury, poisoning and procedural complications | 1590 | 0.59(0.56, 0.62) | 0.62(0.6, 0.64) | 421.82 | −0.7(−0.77) | 0.62(0.59) |
| psychiatric disorders | 952 | 0.59(0.55, 0.63) | 0.6(0.57, 0.64) | 265.47 | −0.73(−0.82) | 0.6(0.57) |
| musculoskeletal and connective tissue disorders | 778 | 0.52(0.48, 0.56) | 0.53(0.49, 0.57) | 339.33 | −0.91(−1.01) | 0.53(0.5) |
| eye disorders | 285 | 0.51(0.45, 0.57) | 0.51(0.45, 0.57) | 134.49 | −0.96(−1.13) | 0.51(0.47) |
| infections and infestations | 531 | 0.35(0.32, 0.38) | 0.36(0.33, 0.39) | 622.83 | −1.46(−1.58) | 0.36(0.34) |
| neoplasms benign, malignant and unspecified (incl cysts and polyps) | 259 | 0.35(0.31, 0.39) | 0.35(0.31, 0.39) | 317.34 | −1.51(−1.68) | 0.35(0.32) |
| surgical and medical procedures | 124 | 0.33(0.28, 0.39) | 0.33(0.28, 0.39) | 169.5 | −1.59(−1.85) | 0.33(0.29) |

*Indicating statistical significance

Abbreviations: ROR, reporting odds ratio; PRR, proportional reporting ratio; CI, confidence interval; χ2, chi-squared; IC, Information Component.

Our analysis revealed a significant number of ADEs associated with spironolactone, with 8,566 reports identifying the drug as the primary suspected cause. The occurrence of these ADEs was more prevalent in females, and the incidence was higher in the elderly population, largely due to its pronounced anti-androgenic effects in treating conditions such as acne, hirsutism, and androgenic alopecia in women [38,39]. Additionally, spironolactone is frequently utilized in feminizing hormone therapy for individuals transitioning from male to female [40]. Conversely, when used by males, spironolactone is more likely to lead to feminizing side effects, such as gynecomastia, and its anti-androgenic activity is less potent in men, which restricts its use for male-related conditions [41]. It is important to highlight that the majority of ADE reports (56.35%) were submitted by healthcare providers rather than patients. This can be attributed to the expertise and clinical experience of medical professionals, which allows them to more effectively detect and assess ADEs. Furthermore, they have greater access to streamlined reporting mechanisms and systems [42]. Additionally, the predominance of reports originating from the United States (28.50%) can be attributed to the higher awareness of drug safety reporting among healthcare professionals and consumers in the country, as well as their increased participation in the reporting process [43]. The large drug market in the United States, coupled with extensive drug use, likely contributes to the higher volume of ADE reports [44]. The most frequent adverse outcome is hospitalization, which is noteworthy given that ADEs contribute significantly to emergency admissions, particularly among the elderly.

**Table 5. The top 30 signal strength of adverse events of spironolactone ranked by ROR at the PTs level in FAERS database.**

| System Organ Class (SOC) | Preferred Term (PT) | Case Reports | ROR(95% CI) | PRR(95% CI) | χ2 | IC(IC025) | EBG-M(EBGM05) |
|---|---|---|---|---|---|---|---|
| reproductive system and breast disorders | endometriosis male | 7 | 13615.84(1675.1, 110674.81) | 13612.3(1671.59, 110849.54) | 11909.01 | 10.73(9.36) | 1702.41(294.87) |
| congenital, familial and genetic disorders | 5-alpha-reductase deficiency | 5 | 1620.81(494.62, 5311.25) | 1620.51(490.24, 5356.66) | 4414.13 | 9.79(8.37) | 884.37(327.59) |
| endocrine disorders | secondary sexual characteristics absence | 3 | 729.31(193.47, 2749.24) | 729.23(192.33, 2764.98) | 1586.69 | 9.05(7.42) | 530.62(174.81) |
| congenital, familial and genetic disorders | bulbospinal muscular atrophy congenital | 6 | 402.42(167.06, 969.36) | 402.33(166.55, 971.92) | 1990.25 | 8.38(7.21) | 333.53(159.83) |
| investigations | blood aldosterone abnormal | 3 | 388.97(112.6, 1343.67) | 388.92(113.13, 1336.99) | 967.31 | 8.34(6.79) | 324.27(114.93) |
| neoplasms benign, malignant and unspecified (incl cysts and polyps) | double hit lymphoma | 5 | 243.12(95.95, 616.06) | 243.08(96.76, 610.69) | 1071.47 | 7.76(6.53) | 216.18(99.3) |
| reproductive system and breast disorders | female sexual arousal disorder | 8 | 207.49(100.08, 430.16) | 207.43(100.44, 428.37) | 1485.04 | 7.55(6.56) | 187.53(101.89) |
| congenital, familial and genetic disorders | hypocalvaria | 10 | 206.95(107.82, 397.24) | 206.87(108.34, 395) | 1851.79 | 7.55(6.65) | 187.08(108.41) |
| psychiatric disorders | gender dysphoria | 7 | 206.3(94.65, 449.68) | 206.25(94.17, 451.73) | 1292.66 | 7.54(6.49) | 186.57(97.2) |
| congenital, familial and genetic disorders | genitalia external ambiguous | 14 | 180.39(104.31, 311.94) | 180.3(104.15, 312.13) | 2284.42 | 7.37(6.6) | 165.08(104.39) |
| congenital, familial and genetic disorders | congenital musculoskeletal disorder | 3 | 132.6(41.17, 427.08) | 132.59(40.91, 429.77) | 366.78 | 6.96(5.48) | 124.19(46.67) |
| reproductive system and breast disorders | asthenospermia | 4 | 131.86(47.89, 363.02) | 131.84(47.58, 365.33) | 486.41 | 6.95(5.64) | 123.53(52.94) |
| investigations | blood aldosterone increased | 8 | 103.74(50.94, 211.29) | 103.71(51.21, 210.02) | 772.58 | 6.62(5.65) | 98.51(54.33) |
| blood and lymphatic system disorders | spur cell anaemia | 3 | 97.24(30.5, 310.07) | 97.23(30.59, 309.04) | 272.12 | 6.53(5.08) | 92.65(35.11) |
| investigations | electrocardiogram t wave peaked | 16 | 94.05(56.94, 155.36) | 94(56.47, 156.47) | 1404.29 | 6.49(5.78) | 89.71(58.95) |
| investigations | electrocardiogram t wave amplitude increased | 4 | 93.73(34.36, 255.65) | 93.72(34.49, 254.66) | 350.04 | 6.48(5.18) | 89.45(38.63) |
| metabolism and nutrition disorders | hyperkalaemia | 1308 | 90.35(85.37, 95.62) | 86.01(81.1, 91.22) | 105303.2 | 6.36(6.28) | 82.41(78.59) |
| reproductive system and breast disorders | oligospermia | 4 | 85.49(31.41, 232.71) | 85.48(31.46, 232.27) | 319.9 | 6.36(5.06) | 81.92(35.44) |
| skin and subcutaneous tissue disorders | urticaria cholinergic | 3 | 84.56(26.61, 268.66) | 84.55(26.6, 268.74) | 237.36 | 6.34(4.89) | 81.07(30.82) |
| skin and subcutaneous tissue disorders | acute cutaneous lupus erythematosus | 5 | 81.04(33.12, 198.27) | 81.03(32.89, 199.62) | 379.38 | 6.28(5.1) | 77.82(36.81) |
| injury, poisoning and procedural complications | therapeutic drug monitoring analysis not performed | 11 | 80.15(43.85, 146.5) | 80.12(43.64, 147.1) | 825.4 | 6.27(5.43) | 76.98(46.48) |
| reproductive system and breast disorders | spermatogenesis abnormal | 3 | 72.93(23.03, 230.95) | 72.92(22.94, 231.77) | 205.12 | 6.14(4.69) | 70.32(26.81) |

*(Continued)*

**Table 5.** (Continued)

| System Organ Class (SOC) | Preferred Term (PT) | Case Reports | ROR(95% CI) | PRR(95% CI) | χ2 | IC(IC025) | EBG-M(EBGM05) |
|---|---|---|---|---|---|---|---|
| neoplasms benign, malignant and unspecified (incl cysts and polyps) | hormone receptor positive breast cancer | 7 | 69.12(32.52, 146.9) | 69.1(32.81, 145.53) | 453.67 | 6.06(5.04) | 66.76(35.53) |
| metabolism and nutrition disorders | hypoosmolar state | 4 | 67.65(24.96, 183.33) | 67.64(24.89, 183.79) | 253.79 | 6.03(4.74) | 65.4(28.4) |
| congenital, familial and genetic disorders | angiotensin converting enzyme inhibitor foetopathy | 3 | 66.3(20.98, 209.56) | 66.29(20.86, 210.7) | 186.57 | 6(4.56) | 64.14(24.49) |
| investigations | renin increased | 8 | 65.66(32.45, 132.84) | 65.64(32.41, 132.93) | 492.62 | 5.99(5.03) | 63.53(35.23) |
| metabolism and nutrition disorders | neonatal hyponatraemia | 3 | 52.56(16.7, 165.48) | 52.56(16.54, 167.06) | 147.74 | 5.68(4.24) | 51.2(19.61) |
| investigations | blood pressure orthostatic | 4 | 51.86(19.21, 140) | 51.86(19.09, 140.91) | 194.32 | 5.66(4.37) | 50.54(22.02) |
| congenital, familial and genetic disorders | gastrointestinal malformation | 5 | 50.65(20.84, 123.09) | 50.64(20.96, 122.33) | 237.13 | 5.63(4.45) | 49.38(23.49) |
| reproductive system and breast disorders | nipple pain | 42 | 46.19(34, 62.74) | 46.12(33.71, 63.11) | 1810.96 | 5.49(5.06) | 45.07(34.88) |

Abbreviations: ROR, reporting odds ratio; PRR, proportional reporting ratio; CI, confidence interval; χ2, chi-squared; IC, Information Component.

This study identified several ADEs linked to spironolactone, including congenital, familial, and genetic disorders, ear and labyrinth disorders, as well as pregnancy, puerperium, and perinatal conditions, none of which are mentioned in the drug's labeling. This underscores the need for more comprehensive drug labeling to better reflect potential ADEs. Conversely, more commonly reported ADEs, such as hyperkalemia and nipple pain, align with existing data, reinforcing their clinical significance. While less frequent, events like male endometriosis, 5-alpha-reductase deficiency, congenital bulbospinal muscular atrophy, and double-hit lymphoma showed strong signal strength, highlighting their potential clinical relevance and the need for further investigation.

Endometriosis has been extensively researched in women, though its exact etiology remains unclear. In rare instances, endometriosis can also occur in men, with the most common sites being the bladder, lower abdominal wall, and inguinal region [45]. Previous theories suggest that factors such as prolonged estrogen therapy [46], cirrhosis [47], or chronic surgical inflammation [48] may contribute to the development of male endometriosis. The patient in question has a history of cirrhosis and has used spironolactone, which could potentially alter hormonal levels and interact with reactive or metabolic processes at the site of prior surgery [49]. 5-alpha-reductase deficiency is an exceptionally rare condition that leads to male sexual development disorders (46, XY karyotype). This enzyme plays a crucial role in converting testosterone into its more biologically active form, 5-alpha-dihydrotestosterone, which is essential for normal masculinization during fetal development. In the absence of this enzyme, the process of masculinization is disrupted, often resulting in infants with ambiguous or predominantly female characteristics [50]. A cohort study of 104 Thai children with unclear genitalia, conducted at Siriraj Hospital, identified only 4 cases of this disorder [51]. Furthermore, a case report documents the first instance of genital abnormalities following prenatal exposure to spironolactone during the first 8 weeks of pregnancy [52]. The affected child presented with male genital anomalies due to 5-alpha-reductase deficiency. Given its rarity, 5-alpha-reductase deficiency is not listed in the prescribing information for spironolactone, highlighting the importance for clinicians to remain vigilant.

Spinal bulbar muscular atrophy (SBMA), a genetic neuromuscular disorder, is a genetic neuromuscular disorder resulting from a trinucleotide repeat expansion in the androgen receptor gene. This condition primarily affects lower motor neurons and skeletal muscles, leading to muscle weakness [53]. Studies have demonstrated that the progression of SBMA is influenced by serum testosterone levels, with spironolactone inhibiting 17α-hydroxylase activity and cytochrome P-450 content, thus effectively decreasing testosterone synthesis. Additionally, spironolactone competes with testosterone for binding to the androgen receptor [54]. As a result, spironolactone could potentially contribute to the clinical manifestation of SBMA in certain individuals [55]. Double-hit lymphoma (DHL), a subtype of high-grade B-cell lymphoma characterized by rearrangements of MYC and BCL2 and/or BCL6 genes, represents an aggressive form of large B-cell lymphoma [56]. It is observed in 5% to 8% of instances and is linked to a poor prognosis despite treatment with R-CHOP chemotherapy [57]. A case report has described male breast involvement in DHL, with the underlying cause still unclear, though it is hypothesized to be linked to spironolactone therapy [58]. Additional investigations are required to examine the underlying mechanisms driving its occurrence.

It is essential to recognize several limitations in this research, some of which are inherent to the FAERS database. First, while disproportionate analysis offers statistically significant signal strength, it cannot establish a definitive causal link between drug exposure and ADEs. Second, the FAERS database lacks detailed clinical context, such as patient comorbidities (e.g., cirrhosis, hormone therapy history) and concomitant medications (e.g., antiandrogens, diuretics), which may independently influence ADEs or interact with spironolactone. Additionally, the FAERS database does not provide information on the actual incidence of ADEs, as the total number of drug exposures (denominator) remains unknown [59]. Furthermore, duplicate entries and missing data from various sources may introduce bias into the findings. Consequently, further prospective studies and clinical trials are essential to validate the causal relationship between spironolactone and these ADEs and to enhance the integration of such data within the FAERS database.

## 5. Conclusion

In conclusion, this study performed a comprehensive assessment of spironolactone's safety profile using the FAERS database. While certain adverse reactions, such as male endometriosis, 5-alpha-reductase deficiency, congenital bulbospinal muscular atrophy, and double-hit lymphoma, are rare, they represent significant signals that warrant further investigation.

## Supporting information

**S1 Table. Application Example of Signal Detection Methods for "Endometriosis Male".**
(DOCX)

## Acknowledgments

The study employed the publicly accessible FAERS database provided by the FDA. It is important to clarify that the data, findings, and interpretations in this study do not represent the views or positions of the FDA.

## Author contributions

**Conceptualization:** Ru Chen.

**Formal analysis:** Chaoning Wen, Lixian Li.

**Funding acquisition:** Chaoning Wen.

**Investigation:** Guowei Lin, Zhuopin Li.

**Methodology:** Chaoning Wen.

**Project administration:** Xiangshan Yan.

**Resources:** Guowei Lin, Ru Chen, Zhuopin Li.

**Software:** Xiangshan Yan.

**Supervision:** Guowei Lin, Lixian Li.

**Validation:** Zhuopin Li.

**Visualization:** Xiangshan Yan.

**Writing – original draft:** Lixian Li.

**Writing – review & editing:** Ru Chen.

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
