## [Decision Letter · Decision Letter 0]

4 Jul 2025

Thank you for submitting your manuscript to PLOS ONE. After careful consideration, we feel that it has merit but does not fully meet PLOS ONE’s publication criteria as it currently stands. Therefore, we invite you to submit a revised version of the manuscript that addresses the points raised during the review process.

We look forward to receiving your revised manuscript.

Kind regards,

Eyob Alemayehu Gebreyohannes, PhD

Academic Editor

PLOS ONE

Journal Requirements:

2. We note that Figure 3 in your submission contain map images which may be copyrighted. All PLOS content is published under the Creative Commons Attribution License (CC BY 4.0), which means that the manuscript, images, and Supporting Information files will be freely available online, and any third party is permitted to access, download, copy, distribute, and use these materials in any way, even commercially, with proper attribution. For these reasons, we cannot publish previously copyrighted maps or satellite images created using proprietary data, such as Google software (Google Maps, Street View, and Earth). For more information, see our copyright guidelines: http://journals.plos.org/plosone/s/licenses-and-copyright.

1) You may seek permission from the original copyright holder of Figure 3 to publish the content specifically under the CC BY 4.0 license.  

2) If you are unable to obtain permission from the original copyright holder to publish these figures under the CC BY 4.0 license or if the copyright holder’s requirements are incompatible with the CC BY 4.0 license, please either i) remove the figure or ii) supply a replacement figure that complies with the CC BY 4.0 license. Please check copyright information on all replacement figures and update the figure caption with source information. If applicable, please specify in the figure caption text when a figure is similar but not identical to the original image and is therefore for illustrative purposes only.

3.  Please remove your figures from within your manuscript file, leaving only the individual TIFF/EPS image files, uploaded separately. These will be automatically included in the reviewers’ PDF.

**Additional Editor Comments:**

REVIEWER 1

Lin et al entitle "Analyzing Real-world Adverse Events of Spironolactone with the FAERS Database" clearly reported that Spironolactone could be adverse drug for heart failure, hypertension, and hyperaldosteronism.

Why are you using the FAERS database to study this drug, it should be that someone else has done this with this similar database similar methodology, so you can write cite in the INTRODUCTION section about some specific other similar studies, such as recommending a few (It is equivalent to saying that someone else has done this type of research using the FAERS database, and you can use this database to do research related to Spironolactone as well):【1】Wang Y, Zhao B, Yang H, Wan Z. A real-world pharmacovigilance study of FDA adverse event reporting system events for sildenafil. Andrology. 2024 May;12(4):785-792. doi: 10.1111/andr.13533. Epub 2023 Sep 19. PMID: 37724699.【2】Zhao B, Fu Y, Cui S, Chen X, Liu S, Luo L. A real-world disproportionality analysis of Everolimus: data mining of the public version of FDA adverse event reporting system. Front Pharmacol. 2024 Mar 12;15:1333662. doi: 10.3389/fphar.2024.1333662. PMID: 38533254; PMCID: PMC10964017.【3】Yang H, Wan Z, Chen M, Zhang X, Cui W, Zhao B. A real-world data analysis of topotecan in the FDA Adverse Event Reporting System (FAERS) database. Expert Opin Drug Metab Toxicol. 2023 Apr;19(4):217-223. doi: 10.1080/17425255.2023.2219390. Epub 2023 May 30. PMID: 37243615.【4】Zhao B, Zhang X, Chen M, Wang Y. A real-world data analysis of acetylsalicylic acid in FDA Adverse Event Reporting System (FAERS) database. Expert Opin Drug Metab Toxicol. 2023 Jan-Jun;19(6):381-387. doi: 10.1080/17425255.2023.2235267. Epub 2023 Jul 12. PMID: 37421631.【5】Li, Jie, Zhao, Bin, Zhu, YongQing, Wu, Jibiao, Vitreoretinal Traction Syndrome, Nitrituria and Human Epidermal Growth Factor Receptor Negative Might Occur in the Aromatase-Inhibitor Anastrozole Treatment, International Journal of Clinical Practice, 2024, 5132916, 9 pages, 2024. https://doi.org/10.1155/2024/5132916【6】Zhong, C., Zheng, Q., Zhao, B., & Ren, T. (2024). A real-world pharmacovigilance study using disproportionality analysis of United States Food and Drug Administration Adverse Event Reporting System events for vinca alkaloids: comparing vinorelbine and Vincristine. Expert Opinion on Drug Safety, 23(11), 1427–1437. https://doi.org/10.1080/14740338.2024.2410436

Besides,

I have general suggestion is:

1.The manuscript mentions the use of ROR, PRR, BCPNN, and EBGM for signal detection. However, the thresholds for significance (e.g., ROR ≥3, PRR ≥2) are not explicitly justified in the context of this study. Please briefly cite relevant literature or guidelines supporting these thresholds to enhance methodological transparency.

2.The Data Availability section states, "The dataset [...] is available from the corresponding author upon request," but the response to the FAERS data availability question on page 5 states, "All relevant data are within the manuscript and its Supporting Information files." Clarify this discrepancy and ensure alignment with PLOS ONE’s data policy (e.g., specify whether raw FAERS data are publicly accessible or require request).

3.Figures 1-3 and Tables 1-5 are referenced in the text but lack captions in the submitted manuscript (e.g., "Fig 1: The flow diagram..." appears only in the text layer). Ensure all figures and tables include descriptive captions in the main manuscript file for clarity.

4.While rare ADEs like "endometriosis male" (n=7) are highlighted, the discussion could benefit from addressing potential confounding factors (e.g., comorbidities, concomitant medications) that might influence these signals. A brief acknowledgment of these limitations would strengthen the interpretation.

5.The ethics statement on page 3 notes that ethical approval was not required. However, the FAERS database includes patient-reported data. Please confirm compliance with FAERS data usage policies and clarify whether patient identifiers were anonymized in accordance with ethical standards.

6.Some references lack consistent formatting (e.g., missing DOIs, inconsistent journal abbreviations). For example, reference 33 (Greenblatt et al., 1973) does not include a DOI. Ensure all references adhere to PLOS ONE’s formatting guidelines.

7.Terms like "SBMA" (spinal bulbar muscular atrophy) and "DHL" (double-hit lymphoma) are introduced in the Discussion without prior definition. Define abbreviations upon first mention to improve readability for non-specialist readers.

REVIEWER 2

The study addresses an important and timely issue in pharmacovigilance by leveraging the FAERS database to explore the safety profile of spironolactone, including the detection of rare and potentially novel adverse events. The methodology is robust, and the findings are potentially impactful for clinical and regulatory audiences.

However, there are some issues which are to be clarified:

1. While you’ve included the key formulas and thresholds used for signal detection, I recommend expanding on how these were applied in practice. For instance, did all four criteria need to be met to consider a signal significant? How were discordant results across methods handled?

A worked example or supplementary figure might be helpful to illustrate the decision-making process.

2. Some of the newly detected signals (e.g., male endometriosis, 5-alpha-reductase deficiency) are quite rare and biologically complex. Adding a brief subsection on the clinical plausibility and supporting literature (if any) would be helpful.

3. The description of data cleaning and deduplication could be strengthened by including a brief example—for instance, how duplicate reports were identified using CASEID and ISR, or how multiple versions of the same case were handled. This would improve methodological clarity and reproducibility.

4. Abbreviations are used in tables and flow chart. Please provide the explanation for those abbreviations below the tables/figures.

ADDITIONAL COMMENTS FROM THE EDITOR

Abstract: Why are only ROR values only reported. The ROR values you reported are also very large. An explanation is needed regarding this.

General: Do adverse event, adverse effect, adverse reaction, and adverse drug event, adverse drug reaction mean the same thing? If so, I suggest you pick one (e.g., ADR) and use it consistently to avoid potential confusion. Also be consistent when using abbreviations.

Methods: The following text belongs in the data analysis, not data collection, section: “The data was analyzed using R software (Version 4.3.1, https://www.r-project.org/).”

Results: The ROR, PRR, EBGM, and IC values appear very high. “Additionally, ADEs such as male endometriosis (n = 7; ROR 13,615.84, PRR 13,612.3, IC 10.73, EBGM 1,702.41), 5-alpha-reductase deficiency (n = 5; ROR 1,620.81, PRR 1,620.51, IC 9.79, EBGM 884.37), congenital bulbospinal muscular atrophy (n = 6; ROR 402.42, PRR 402.33, IC 8.38, EBGM 333.53), and double-hit lymphoma (n = 5; ROR 243.12, PRR 243.08, IC 7.76, EBGM 216.18).” It may be useful to expand your methods to describe how each of these measures are calculated.

Reviewers' comments:

Reviewer's Responses to Questions

**Comments to the Author**

1. Is the manuscript technically sound, and do the data support the conclusions?

Reviewer #1: Yes

Reviewer #2: Yes

2. Has the statistical analysis been performed appropriately and rigorously?

Reviewer #1: Yes

Reviewer #2: I Don't Know

3. Have the authors made all data underlying the findings in their manuscript fully available?

Reviewer #1: Yes

Reviewer #2: Yes

4. Is the manuscript presented in an intelligible fashion and written in standard English?

Reviewer #1: Yes

Reviewer #2: Yes

Reviewer #1: Lin et al entitle "Analyzing Real-world Adverse Events of Spironolactone with the FAERS Database" clearly reported that Spironolactone could be adverse drug for heart failure, hypertension, and hyperaldosteronism.

Why are you using the FAERS database to study this drug, it should be that someone else has done this with this similar database similar methodology, so you can write cite in the INTRODUCTION section about some specific other similar studies, such as recommending a few (It is equivalent to saying that someone else has done this type of research using the FAERS database, and you can use this database to do research related to Spironolactone as well):【1】Wang Y, Zhao B, Yang H, Wan Z. A real-world pharmacovigilance study of FDA adverse event reporting system events for sildenafil. Andrology. 2024 May;12(4):785-792. doi: 10.1111/andr.13533. Epub 2023 Sep 19. PMID: 37724699.【2】Zhao B, Fu Y, Cui S, Chen X, Liu S, Luo L. A real-world disproportionality analysis of Everolimus: data mining of the public version of FDA adverse event reporting system. Front Pharmacol. 2024 Mar 12;15:1333662. doi: 10.3389/fphar.2024.1333662. PMID: 38533254; PMCID: PMC10964017.【3】Yang H, Wan Z, Chen M, Zhang X, Cui W, Zhao B. A real-world data analysis of topotecan in the FDA Adverse Event Reporting System (FAERS) database. Expert Opin Drug Metab Toxicol. 2023 Apr;19(4):217-223. doi: 10.1080/17425255.2023.2219390. Epub 2023 May 30. PMID: 37243615.【4】Zhao B, Zhang X, Chen M, Wang Y. A real-world data analysis of acetylsalicylic acid in FDA Adverse Event Reporting System (FAERS) database. Expert Opin Drug Metab Toxicol. 2023 Jan-Jun;19(6):381-387. doi: 10.1080/17425255.2023.2235267. Epub 2023 Jul 12. PMID: 37421631.【5】Li, Jie, Zhao, Bin, Zhu, YongQing, Wu, Jibiao, Vitreoretinal Traction Syndrome, Nitrituria and Human Epidermal Growth Factor Receptor Negative Might Occur in the Aromatase-Inhibitor Anastrozole Treatment, International Journal of Clinical Practice, 2024, 5132916, 9 pages, 2024. https://doi.org/10.1155/2024/5132916【6】Zhong, C., Zheng, Q., Zhao, B., & Ren, T. (2024). A real-world pharmacovigilance study using disproportionality analysis of United States Food and Drug Administration Adverse Event Reporting System events for vinca alkaloids: comparing vinorelbine and Vincristine. Expert Opinion on Drug Safety, 23(11), 1427–1437. https://doi.org/10.1080/14740338.2024.2410436

Besides,

I have general suggestion is:

1.The manuscript mentions the use of ROR, PRR, BCPNN, and EBGM for signal detection. However, the thresholds for significance (e.g., ROR ≥3, PRR ≥2) are not explicitly justified in the context of this study. Please briefly cite relevant literature or guidelines supporting these thresholds to enhance methodological transparency.

2.The Data Availability section states, "The dataset [...] is available from the corresponding author upon request," but the response to the FAERS data availability question on page 5 states, "All relevant data are within the manuscript and its Supporting Information files." Clarify this discrepancy and ensure alignment with PLOS ONE’s data policy (e.g., specify whether raw FAERS data are publicly accessible or require request).

3.Figures 1-3 and Tables 1-5 are referenced in the text but lack captions in the submitted manuscript (e.g., "Fig 1: The flow diagram..." appears only in the text layer). Ensure all figures and tables include descriptive captions in the main manuscript file for clarity.

4.While rare ADEs like "endometriosis male" (n=7) are highlighted, the discussion could benefit from addressing potential confounding factors (e.g., comorbidities, concomitant medications) that might influence these signals. A brief acknowledgment of these limitations would strengthen the interpretation.

5.The ethics statement on page 3 notes that ethical approval was not required. However, the FAERS database includes patient-reported data. Please confirm compliance with FAERS data usage policies and clarify whether patient identifiers were anonymized in accordance with ethical standards.

6.Some references lack consistent formatting (e.g., missing DOIs, inconsistent journal abbreviations). For example, reference 33 (Greenblatt et al., 1973) does not include a DOI. Ensure all references adhere to PLOS ONE’s formatting guidelines.

7.Terms like "SBMA" (spinal bulbar muscular atrophy) and "DHL" (double-hit lymphoma) are introduced in the Discussion without prior definition. Define abbreviations upon first mention to improve readability for non-specialist readers.

Reviewer #2: The study addresses an important and timely issue in pharmacovigilance by leveraging the FAERS database to explore the safety profile of spironolactone, including the detection of rare and potentially novel adverse events. The methodology is robust, and the findings are potentially impactful for clinical and regulatory audiences.

However, there are some issues which are to be clarified:

1. While you’ve included the key formulas and thresholds used for signal detection, I recommend expanding on how these were applied in practice. For instance, did all four criteria need to be met to consider a signal significant? How were discordant results across methods handled?

A worked example or supplementary figure might be helpful to illustrate the decision-making process.

2. Some of the newly detected signals (e.g., male endometriosis, 5-alpha-reductase deficiency) are quite rare and biologically complex. Adding a brief subsection on the clinical plausibility and supporting literature (if any) would be helpful.

3. The description of data cleaning and deduplication could be strengthened by including a brief example—for instance, how duplicate reports were identified using CASEID and ISR, or how multiple versions of the same case were handled. This would improve methodological clarity and reproducibility.

4. Abbreviations are used in tables and flow chart. Please provide the explanation for those abbreviations below the tables/figures.

**Do you want your identity to be public for this peer review?** For information about this choice, including consent withdrawal, please see our Privacy Policy

Reviewer #1: No

Reviewer #2: **Yes: ** Dr. Meenalotchini Prakash Gurunthalingam

---

## [Author Response · Author response to Decision Letter 1]

10 Jul 2025

Dear Editors and Reviewers:

On behalf of my co-authors, we thank you very much for giving us an opportunity to revise our manuscript, and we appreciate the editor and reviewers very much for their positive and constructive comments and suggestions on our manuscript entitled “Analyzing Real-world Adverse Events of Spironolactone with the FAERS Database”. Those comments are all valuable and very helpful for revising and improving our paper, as well as the important guiding significance to our research. We have studied comments carefully and have made corrections which we hope will meet with approval. The main corrections in the paper and the responses to the reviewers’ comments are as follows:

Responds to the reviewers’ comments:

Journal Requirements:

Response: We have carefully revised the manuscript to fully comply with PLOS ONE's style requirements, including proper file naming conventions as specified in the provided templates. All structural elements (such as section headings, figure/table formats, and reference citations) have been formatted according to the journal's guidelines, and supporting information files have been named appropriately. The document now adheres to the specified styling for titles, authors, affiliations, and supplementary materials, ensuring alignment with PLOS ONE's submission standards.

2) We note that Figure 3 in your submission contain map images which may be copyrighted. All PLOS content is published under the Creative Commons Attribution License (CC BY 4.0), which means that the manuscript, images, and Supporting Information files will be freely available online, and any third party is permitted to access, download, copy, distribute, and use these materials in any way, even commercially, with proper attribution. For these reasons, we cannot publish previously copyrighted maps or satellite images created using proprietary data, such as Google software (Google Maps, Street View, and Earth). For more information, see our copyright guidelines: http://journals.plos.org/plosone/s/licenses-and-copyright.We require you to either (1) present written permission from the copyright holder to publish these figures specifically under the CC BY 4.0 license, or (2) remove the figures from your submission: You may seek permission from the original copyright holder of Figure 3 to publish the content specifically under the CC BY 4.0 license. We recommend that you contact the original copyright holder with the Content Permission Form (http://journals.plos.org/plosone/s/file?id=7c09/content-permission-form.pdf) and the following text: “I request permission for the open-access journal PLOS ONE to publish XXX under the Creative Commons Attribution License (CCAL) CC BY 4.0 (http://creativecommons.org/licenses/by/4.0/). Please be aware that this license allows unrestricted use and distribution, even commercially, by third parties. Please reply and provide explicit written permission to publish XXX under a CC BY license and complete the attached form.” Please upload the completed Content Permission Form or other proof of granted permissions as an ""Other"" file with your submission. In the figure caption of the copyrighted figure, please include the following text: “Reprinted from [ref] under a CC BY license, with permission from [name of publisher], original copyright [original copyright year].”

Response: Thank you for bringing this to our attention. We have promptly removed Figure 3 from the manuscript as requested, and confirm that this modification does not impact the scientific integrity or narrative flow of the original content. All references to the figure within the text have been carefully adjusted to maintain coherence. We appreciate your guidance in ensuring compliance with PLOS ONE's licensing and copyright standards, and stand ready to provide any further documentation or revisions as needed.

3) Please remove your figures from within your manuscript file, leaving only the individual TIFF/EPS image files, uploaded separately. These will be automatically included in the reviewers’ PDF.

Response: We have followed your instructions to remove all figures from the main manuscript file. The figures have been saved as individual TIFF/EPS image files and will be uploaded separately as required. This ensures they will be automatically included in the reviewers’ PDF, while the main document now only contains text and figure citations for clarity.

4) Please include captions for your Supporting Information files at the end of your manuscript, and update any in-text citations to match accordingly. Please see our Supporting Information guidelines for more information: http://journals.plos.org/plosone/s/supporting-information.

Response: We have added captions for the Supporting Information files at the end of the manuscript and updated the in-text citations accordingly. In the Methods section, the corresponding in-text citation has been revised to "S1 Table" to match this caption.

Reviewer 1

Lin et al entitle "Analyzing Real-world Adverse Events of Spironolactone with the FAERS Database" clearly reported that Spironolactone could be adverse drug for heart failure, hypertension, and hyperaldosteronism.

1) Why are you using the FAERS database to study this drug, it should be that someone else has done this with this similar database similar methodology, so you can write cite in the INTRODUCTION section about some specific other similar studies, such as recommending a few (It is equivalent to saying that someone else has done this type of research using the FAERS database, and you can use this database to do research related to Spironolactone as well):【1】Wang Y, Zhao B, Yang H, Wan Z. A real-world pharmacovigilance study of FDA adverse event reporting system events for sildenafil. Andrology. 2024 May;12(4):785-792. doi: 10.1111/andr.13533. Epub 2023 Sep 19. PMID: 37724699.【2】Zhao B, Fu Y, Cui S, Chen X, Liu S, Luo L. A real-world disproportionality analysis of Everolimus: data mining of the public version of FDA adverse event reporting system. Front Pharmacol. 2024 Mar 12;15:1333662. doi: 10.3389/fphar.2024.1333662. PMID: 38533254; PMCID: PMC10964017.【3】Yang H, Wan Z, Chen M, Zhang X, Cui W, Zhao B. A real-world data analysis of topotecan in the FDA Adverse Event Reporting System (FAERS) database. Expert Opin Drug Metab Toxicol. 2023 Apr;19(4):217-223. doi: 10.1080/17425255.2023.2219390. Epub 2023 May 30. PMID: 37243615.【4】Zhao B, Zhang X, Chen M, Wang Y. A real-world data analysis of acetylsalicylic acid in FDA Adverse Event Reporting System (FAERS) database. Expert Opin Drug Metab Toxicol. 2023 Jan-Jun;19(6):381-387. doi: 10.1080/17425255.2023.2235267. Epub 2023 Jul 12. PMID: 37421631.【5】Li, Jie, Zhao, Bin, Zhu, YongQing, Wu, Jibiao, Vitreoretinal Traction Syndrome, Nitrituria and Human Epidermal Growth Factor Receptor Negative Might Occur in the Aromatase-Inhibitor Anastrozole Treatment, International Journal of Clinical Practice, 2024, 5132916, 9 pages, 2024. https://doi.org/10.1155/2024/5132916【6】Zhong, C., Zheng, Q., Zhao, B., & Ren, T. (2024). A real-world pharmacovigilance study using disproportionality analysis of United States Food and Drug Administration Adverse Event Reporting System events for vinca alkaloids: comparing vinorelbine and Vincristine. Expert Opinion on Drug Safety, 23(11), 1427–1437. https://doi.org/10.1080/14740338.2024.2410436.

Response: Thank you for the feedback. We have incorporated the suggested references in the Introduction section to contextualize the use of the FAERS database, aligning with prior studies that have employed similar methodologies. This addition strengthens the rationale for our approach and highlights the database's utility in pharmacovigilance research. These revisions have been incorporated into the revised Introduction section (paragraph 3).

2) The manuscript mentions the use of ROR, PRR, BCPNN, and EBGM for signal detection. However, the thresholds for significance (e.g., ROR ≥3, PRR ≥2) are not explicitly justified in the context of this study. Please briefly cite relevant literature or guidelines supporting these thresholds to enhance methodological transparency.

Response: Thank you for pointing out the need to justify the thresholds for signal detection methods. We have addressed this by supplementing the methodology with citations to relevant literature. The thresholds were selected based on established standards in pharmacovigilance: ROR ≥ 3 and 95% CI lower limit > 1 follow Rothman et al. (2004), PRR ≥ 2 aligns with Evans et al. (2001), BCPNN uses IC025 > 0 as recommended by Bate et al. (1998), and EBGM05 > 2 follows Dumouchel (1999). These references now appear in the manuscript to validate the criteria, ensuring methodological transparency and consistency with prior research in the field. These revisions have been incorporated into the revised Methods section 2.2 (paragraph 2).

3) The Data Availability section states, "The dataset [...] is available from the corresponding author upon request," but the response to the FAERS data availability question on page 5 states, "All relevant data are within the manuscript and its Supporting Information files." Clarify this discrepancy and ensure alignment with PLOS ONE’s data policy (e.g., specify whether raw FAERS data are publicly accessible or require request).

Response: Thank you for pointing out the discrepancy in the data availability statements. The confusion stems from the distinction between the raw FAERS data and the study-specific processed dataset. The raw FAERS data used in this analysis are publicly accessible via the FDA’s official repository (https://fis.fda.gov/extensions/FPD-QDE-FAERS/FPD-QDE-FAERS.html), as acknowledged in the manuscript. In contrast, the processed dataset (e.g., cleaned reports, annotated variables, and analysis outputs) generated during this study is available from the corresponding author upon reasonable request, as stated in the Data Availability section. To align with PLOS ONE’s data policy, the statement has been revised to clarify: "The raw FAERS data used in this study are publicly available from the U.S. Food and Drug Administration (FDA) (https://fis.fda.gov/extensions/FPD-QDE-FAERS/FPD-QDE-FAERS.html). The processed dataset and analysis scripts generated during the current study are available from the corresponding author upon reasonable request." This revision explicitly differentiates between publicly accessible raw data and study-generated processed data, ensuring transparency and compliance with journal guidelines. These revisions have been incorporated into the revised Availability of data and materials section.

4) Figures 1-3 and Tables 1-5 are referenced in the text but lack captions in the submitted manuscript (e.g., "Fig 1: The flow diagram..." appears only in the text layer). Ensure all figures and tables include descriptive captions in the main manuscript file for clarity.

Response: Thank you for noting the absence of captions for Figures 1-3 and Tables 1-5 in the manuscript. We have addressed this by incorporating descriptive captions for all figures and tables directly within the main document to enhance clarity and compliance with publishing standards. Each figure caption now concisely outlines the workflow or key elements. These revisions ensure that visual and tabular content is self-explanatory and fully integrated with the text, eliminating reliance on in-text descriptions alone and aligning with academic formatting requirements.

5) While rare ADEs like "endometriosis male" (n=7) are highlighted, the discussion could benefit from addressing potential confounding factors (e.g., comorbidities, concomitant medications) that might influence these signals. A brief acknowledgment of these limitations would strengthen the interpretation.

Response: We acknowledge the potential confounding factors influencing rare ADE signals (e.g., male endometriosis, n=7). FAERS data lack detailed documentation of comorbidities (e.g., cirrhosis, hormone therapy history) and concomitant medications (e.g., antiandrogens, diuretics), which may independently contribute to these events or interact with spironolactone. The voluntary reporting nature may introduce selection bias, hindering causal inference. Prospective studies or electronic health record analyses controlling for clinical confounders are needed to validate these signals. We have added relevant content in the discussion section. These revisions have been incorporated into the revised Discussion section (last paragraph).

6) The ethics statement on page 3 notes that ethical approval was not required. However, the FAERS database includes patient-reported data. Please confirm compliance with FAERS data usage policies and clarify whether patient identifiers were anonymized in accordance with ethical standards.

Response: Our study adheres to FAERS data usage policies and ethical standards. The FAERS database publicly provides anonymized adverse event reports, with patient identifiers and sensitive information removed in compliance with FDA regulations. As stated in the ethics statement (Section 12), the research involved secondary analysis of fully anonymized, publicly available data, which does not constitute human subjects research requiring ethical approval. All data used in this study were de-identified prior to access, aligning with both FAERS guidelines and institutional ethical requirements for non-identifiable dataset analysis.

7) Some references lack consistent formatting (e.g., missing DOIs, inconsistent journal abbreviations). For example, reference 33 (Greenblatt et al., 1973) does not include a DOI. Ensure all references adhere to PLOS ONE’s formatting guidelines.

Response: We have standardized all references to comply with PLOS ONE’s formatting guidelines, including adding missing DOIs and ensuring consistent journal abbreviations. For references without DOIs (e.g., older studies), we’ve followed the journal’s requirements for legacy publications.

8) Terms like "SBMA" (spinal bulbar muscular atrophy) and "DHL" (double-hit lymphoma) are introduced in the Discussion without prior definition. Define abbreviations upon first mention to improve readability for non-specialist readers.

Response: We appreciate the feedback on enhancing readability for non-specialist readers. In the revised manuscript, we have ensured that abbreviations such as "SBMA" (spinal bulbar muscular atrophy) and "DHL" (double-hit lymphoma) are defined at their first occurrence in the Discussion section. These revisions have been incorporated into the revised Discussion section (paragraph 5).

Reviewer 2:

The study addresses an important and timely issue in pharmacovigilance by leveraging the FAERS database to explore the safety profile of spironolactone, including the detection of rare and potentially novel adverse events. The methodology is robust, and the findings are potentially impactful for clinical and regulatory audiences.

However, there are some issues which are to be clarified:

1) While you’ve included the key formulas and thresholds used for signal detection, I recommend expanding on how these were applied in practice. For instance, did all four criteria need to be met to consider a signal significant? How were discordant results across methods handled? A worked example or supplementary figure might be helpful to illustrate the decision-making process.

Response: We appreciate the suggestion to elaborate on the practical application of signal detection methods. In this study, a signal was considered significant only when all four statistical methods (PRR, ROR, BCPNN, EBGM) met their respective threshold criteria (e.g., ROR ≥ 3 with 95% CI lower limit > 1, PRR ≥ 2, IC025 > 0, EBGM05 > 2). This multi-algorithm approach was deliberately implemented to minimize false positives by requiring cros

---

## [Decision Letter · Decision Letter 1]

1 Aug 2025

Dear Dr. Lin,

Thank you for submitting your manuscript to PLOS ONE. After careful consideration, we feel that it has merit but does not fully meet PLOS ONE’s publication criteria as it currently stands. Therefore, we invite you to submit a revised version of the manuscript that addresses the points raised during the review process.

We look forward to receiving your revised manuscript.

Kind regards,

Eyob Alemayehu Gebreyohannes, PhD

Academic Editor

PLOS ONE

**Journal Requirements:**

**Additional Editor Comments:**

Introduction: Despite the therapeutic benefits of spironolactone, it is associated with a range of adverse drug event (ADE)… add ‘s’ after ‘adverse drug event’ and ‘ADE’: i.e., “adverse drug events (ADEs)”

In response to the editor’s request for consistency in terminology (ADE, AE, ADR, etc.), you stated that the term "ADE" was used consistently. However, this does not appear to be the case. For example: “The common adverse event (AE) is…”   “Other infrequent AE…” Please revise the manuscript to ensure consistent use of terminology, preferably standardizing to “ADEs” if that is your chosen term.

Additionally, in your response to Reviewer 1’s first comment, you noted that paragraph 3 of the Introduction section was revised accordingly. However, the only visible change is the substitution of “adverse drug reactions” with “ADEs.” Please revisit this section to ensure the reviewer’s comment has been fully addressed.

Regarding Table 2, it appears overly congested, making it difficult to observe which formula corresponds to which measure. I suggest adding appropriate spacing or separating lines to clearly delineate the formulas for each of the four measures: ROR, PRR, BCPNN, and EBGM.

Reviewers' comments:

Reviewer's Responses to Questions

**Comments to the Author**

1. If the authors have adequately addressed your comments raised in a previous round of review and you feel that this manuscript is now acceptable for publication, you may indicate that here to bypass the “Comments to the Author” section, enter your conflict of interest statement in the “Confidential to Editor” section, and submit your "Accept" recommendation.

Reviewer #2: All comments have been addressed

2. Is the manuscript technically sound, and do the data support the conclusions?

Reviewer #2: Yes

3. Has the statistical analysis been performed appropriately and rigorously?

Reviewer #2: Yes

4. Have the authors made all data underlying the findings in their manuscript fully available?

Reviewer #2: Yes

5. Is the manuscript presented in an intelligible fashion and written in standard English?

Reviewer #2: Yes

**Reviewer #2: ** I appreciate that the authors have taken the time to carefully and thoroughly address each of the comments and suggestions I had raised in the previous round of review. Their responses are comprehensive, thoughtful, and demonstrate a clear understanding of the issues highlighted. It is evident that they have made a sincere effort to incorporate the feedback into the revised manuscript, resulting in substantial improvements in both clarity and scientific rigor. The revisions have enhanced the overall quality of the work, and I am satisfied with how my concerns have been handled in the current version of the manuscript.

**Do you want your identity to be public for this peer review?** For information about this choice, including consent withdrawal, please see our Privacy Policy

Reviewer #2: **Yes: ** Dr. G. Meenalotchini

---

## [Author Response · Author response to Decision Letter 2]

3 Aug 2025

Dear Editors and Reviewers:

On behalf of my co-authors, we thank you very much for giving us an opportunity to revise our manuscript, and we appreciate the editor and reviewers very much for their positive and constructive comments and suggestions on our manuscript entitled “Analyzing Real-world Adverse Events of Spironolactone with the FAERS Database”. Those comments are all valuable and very helpful for revising and improving our paper, as well as the important guiding significance to our research. We have studied comments carefully and have made corrections which we hope will meet with approval. The main corrections in the paper and the responses to the reviewers’ comments are as follows:

Responds to the reviewers’ comments:

Journal Requirements:

Response: We have carefully reviewed the specific previously published works recommended in the reviewer comments and confirmed their relevance. Additionally, we have thoroughly checked our reference list to ensure its completeness and accuracy. No irrelevant citations are included, and none of the cited papers have been retracted. All references are current and relevant to the content of the manuscript.

Additional Editor Comments

1) Introduction: Despite the therapeutic benefits of spironolactone, it is associated with a range of adverse drug event (ADE)… add ‘s’ after ‘adverse drug event’ and ‘ADE’: i.e., “adverse drug events (ADEs)”.

Response: Thank you for your thoughtful suggestion. We have revised the relevant content as recommended, changing "adverse drug event (ADE)" to "adverse drug events (ADEs)". Please feel free to review the updated version, and we appreciate your careful attention to detail. These revisions have been incorporated into the revised Introduction section (paragraph 2).

2) In response to the editor’s request for consistency in terminology (ADE, AE, ADR, etc.), you stated that the term "ADE" was used consistently. However, this does not appear to be the case. For example: “The common adverse event (AE) is…” “Other infrequent AE…” Please revise the manuscript to ensure consistent use of terminology, preferably standardizing to “ADEs” if that is your chosen term.

Response: Thank you for pointing out the inconsistency in the use of terminology such as ADE, AE, and ADR in the manuscript. We sincerely appreciate your careful review. Following your suggestion, we have thoroughly revised the entire manuscript to ensure consistent terminology. We have standardized the term to "ADEs" (adverse drug events) throughout, replacing instances where "AE" (adverse event) was used incorrectly, including the examples you mentioned: "The common adverse event (AE) is..." and "Other infrequent AE...". We have double-checked the manuscript to confirm that all relevant terms are now used consistently. Please let us know if there are any further issues that need attention. Thank you again for your valuable feedback.

3) Additionally, in your response to Reviewer 1’s first comment, you noted that paragraph 3 of the Introduction section was revised accordingly. However, the only visible change is the substitution of “adverse drug reactions” with “ADEs.” Please revisit this section to ensure the reviewer’s comment has been fully addressed.

Response: Thank you for your careful attention to this matter. We apologize for the lack of clarity in our previous response. Regarding Reviewer 1’s first comment, which requested the addition of relevant references to strengthen the Introduction, we had indeed addressed this in our revision by incorporating references 19–24 into paragraph 3 of the Introduction. These references were added to provide further supporting evidence for the statements in that section, aligning with the reviewer’s suggestion. The substitution of “adverse drug reactions” with “ADEs” was a separate adjustment for terminology consistency, and we recognize that this may have overshadowed the reference additions in the visible text. We have rechecked the section to confirm that references 19–24 are properly integrated to support the content, and we ensure that Reviewer 1’s comment has now been fully addressed.

4) Regarding Table 2, it appears overly congested, making it difficult to observe which formula corresponds to which measure. I suggest adding appropriate spacing or separating lines to clearly delineate the formulas for each of the four measures: ROR, PRR, BCPNN, and EBGM.

Response: Thank you for your suggestion on Table 2. We have revised it by adding appropriate spacing and separating lines to clearly distinguish the formulas for ROR, PRR, BCPNN, and EBGM. Please review.

We sincerely thank you again for your thoughtful comments and constructive suggestions. We hope that the revised manuscript meets the requirements and expectations. Please do not hesitate to contact us should any further clarification be needed.

With best regards,

Lixian Li

lilixian007@sina.com

---

## [Editor Report · Decision Letter 2]

5 Aug 2025

Analyzing Real-world Adverse Events of Spironolactone with the FAERS Database

PONE-D-25-05600R2

Dear Dr. Li,

We’re pleased to inform you that your manuscript has been judged scientifically suitable for publication and will be formally accepted for publication once it meets all outstanding technical requirements.

Kind regards,

Eyob Alemayehu Gebreyohannes, PhD

Academic Editor

PLOS ONE
---

## [Editor Report · Acceptance letter]

PONE-D-25-05600R2

PLOS ONE

Dear Dr. Li,

I'm pleased to inform you that your manuscript has been deemed suitable for publication in PLOS ONE. Congratulations! Your manuscript is now being handed over to our production team.

Kind regards,

on behalf of

Dr. Eyob Alemayehu Gebreyohannes

Academic Editor

PLOS ONE